# On the temperature stability requirements of free-running Nd:YAG lasers for atmospheric temperature profiling through the rotational Raman technique

José Alex Zenteno-Hernández[1,2], Adolfo Comerón[2*], Federico Dios[2], Alejandro Rodríguez-Gómez[2], Constantino Muñoz-Porcar[2], Michaël Sicard[2,3], Noemi Franco[4], Andreas Behrendt[5], Paolo Di Girolamo[4]

[1]Instituto Nacional de Astrofísica, Óptica y Electrónica (INAOE), 72840 Puebla, Mexico
[2]CommSensLab, Dept. of Signal Theory and Communications, Universitat Politècnica de Catalunya (UPC), 08034 Barcelona, Spain
[3]Laboratoire de l'Atmosphère et des Cyclones (LACy), Université de la Réunion, Saint Denis, 97744, France
[4]Scuola di Ingegneria, Università della Basilicata, 85100 Potenza, Italy
[5]University of Hohenheim, Institute of Physics and Meteorology, 70599 Stuttgart, Germany

*Correspondence to*: Adolfo Comerón (comeron@tsc.upc.edu)

**Abstract.** We assess the temperature stability requirements of unseeded Nd:YAG lasers in lidar systems for atmospheric temperature profiling through the rotational Raman technique. Taking as a reference a system using a seeded laser assumed to emit pulses of negligible spectral width and wavelength-drift free, we estimate first the effect of the pulse spectral widening of the unseeded laser on the output of the interference filters, then we derive the limits of the allowable wavelength drift for a given bias in the temperature measurement that would add to the noise-induced uncertainty. Finally, using spectroscopic data, we relate the allowable wavelength drift to allowable temperature variations of the YAG rod. We find that, in order to keep the bias affecting atmospheric temperature measurements smaller than 1 K, the Nd:YAG rod temperature should also be kept within 1 K.

## 1 Introduction

Profiles of humidity and temperature throughout the troposphere are of the utmost importance to weather forecast (Statement of Guidance for Global Numerical Weather Forecasting (GNWP), 2023). Radiosondes are the staple way to provide such profiles; nevertheless, out of specific measurement campaigns, radiosondes are launched only every 12 hours from most meteorological-service sites; on the other hand, Raman lidars can operate on a 24/7 basis, obtaining time resolutions in the range from minutes to a few hours. Therefore, Raman lidars offer an alternative to retrieve these profiles with relatively high temporal resolution without expending balloons and their payloads. With respect to temperature profiles, the Raman lidar technique is based on the dependence on temperature of the intensity of the atmospheric $N_2$ and $O_2$ rotational Raman lines (Cooney, 1972). In the observed range of atmospheric temperatures, the backscatter cross section corresponding to the Raman lines with low rotational quantum numbers tends to decrease when the temperature increases, while that

corresponding to lines with high rotational quantum numbers has the opposite behavior. Therefore, the power ratio of the backscattered radiation in two spectral regions provides information on the temperature of the backscattering volume.

In the past, the separation of the backscatter of different regions of the pure rotational Raman spectrum has been implemented through the use of diffraction gratings and spatial filtering (Arshinov et al., 1983), and some current systems use a similar technique directing the radiation of different rotational Raman lines to the ends of different optical fibers (Martucci et al., 2021). The current state of the art permits the use of interference filters to do that separation (Vaughan et al., 1993; Behrendt and Reichardt, 2000).

The power ratio of the filter outputs corresponding to the different regions of the spectrum needs to be calibrated to retrieve the atmospheric temperature. This is usually done by comparison with a radiosonde profile, e.g. Hammann et al., 2015, although in-situ calibrations using local light sources (Vaughan et al., 1993), or by constructing a model of the power ratio between the outputs of the filters for high- and low-quantum numbers (Mahagammulla Gamage et al., 2019) are also described in the literature. For the calibration to be valid over long periods, the stability of the system to ensure that the filters' response keeps constant in time is of paramount importance. Modern interference filters can have temperature dependences between 2 pm/ºC and 5 pm/ºC (Johansen et al., 2017) and, being usually kept in a controlled environment, where temperature can be easily stabilized, their response stability is usually not an issue. Of more concern regarding system stability is the laser wavelength. Although some systems use free-running lasers (Di Girolamo et al., 2004; Mahagammulla Gamage et al., 2019), laser frequency stability is achieved in many systems through the use of injection-seeded lasers, where the stability and spectral purity of the seeder is transferred to the host laser (Behrendt and Reichardt, 2000; Behrendt et al., 2002; Behrendt et al., 2004; Lange et al., 2019), which is forced to operate in a single longitudinal mode. As the cost of a free-running laser is much lower than that of a seeded laser for equivalent output energies, the use of the former can be of advantage, provided the cost of achieving the wavelength stability assessed in the present paper does not offset the reduction on laser cost.

In this paper we assume that the receiver is in a well-controlled environment, such that the wavelength drift of the filters can be neglected. We focus henceforth on the wavelength stability requirements for a Nd:YAG non-seeded laser to be used in a lidar measuring atmospheric temperature profiles using the pure rotational Raman spectra of $N_2$ and $O_2$ under the excitation by the third harmonic of the laser fundamental frequency. In section 2, the short-term spectral widening of a free-running frequency tripled Nd:YAG laser is taken into account. Section 3 considers the far more important effect of temperature variations in the YAG rod. Section 4 presents the conclusions of this work.

**2 Effect of short-term emission spectral widening**

We assume that a free-running Nd:YAG laser oscillates in the most intense line, i.e. the R2 → Y3 line (Kushida, 1969) at 1064 nm. This line has a gain full width at half maximum of approximately 5 cm⁻¹ (Verdeyen, 1989; Sato and Taira, 2012). As only the modes closest to the maximum gain can oscillate, the emission linewidth is considerably narrower. We will

assume a typical emission linewidth of the free-running laser of 1 cm$^{-1}$ at 1064 nm (in fact, references (Armandillo et al., 1997; Lumibird, 2018) give a somewhat lower figure of approximately 0.7 cm$^{-1}$), which would result in 3 cm$^{-1}$ at the third harmonic wavelength. Although this linewidth is made of many competing modes, we will assume that its average over a sufficient number of pulses can be modeled by a Gaussian curve (Armandillo et al., 1997). Under the excitation of a widened emission, the Raman backscatter spectrum lines are widened accordingly, therefore we will consider that each line of backscatter differential cross-section $\sigma_i$ will show a profile as a function of frequency given by

$$f_{\sigma_i}(\nu) = \frac{2\sqrt{\ln 2}\,\sigma_i}{\sqrt{\pi}\Delta\nu} \exp\left[-\frac{4\ln 2\left(\nu-\nu_{0i}\right)^2}{\left(\Delta\nu\right)^2}\right],$$
(1)

with $\nu_{0i}$ the central emission-line frequency and $\Delta\nu$ the full width at half maximum of the excitation spectrum.

For the purpose of comparison, we will consider the filter responses like those in Hammann et al., 2015. These filters were specified after a thorough analysis of the system trade-off between sensitivity to temperature difference and statistical-noise induced uncertainty (Radlach et al., 2008; Hammann et al., 2015); for this reason, we expect their characteristics (bandwidth and central-wavelength distance to the laser emission line) be representative of modern Raman temperature lidars using interference filters. Figure 1 shows the widened Cabannes lines and pure rotational Raman backscatter spectrum (in terms of spectral density) of $N_2$ and $O_2$ at sea level (NASA, 1976) with the filters of Hammann et al., 2015 and assuming a central wavelength of the excitation radiation of 354.7133 nm, the 3rd harmonic wavelength of a Nd:YAG laser at a rod temperature of 38 ºC (see section 3). Table 1 shows the wavelengths (in vacuum) and wavenumbers considered. The calculation is based on the spectrum frequencies and intensities used in Zenteno-Hernández et al., 2021, which in turn rely on expressions and spectroscopic data found in Haga clic o pulse aquí para escribir texto.Murphy et al., 2016; Alms et al., 2008; Buldakov et al., 1996; and Long, 2002.

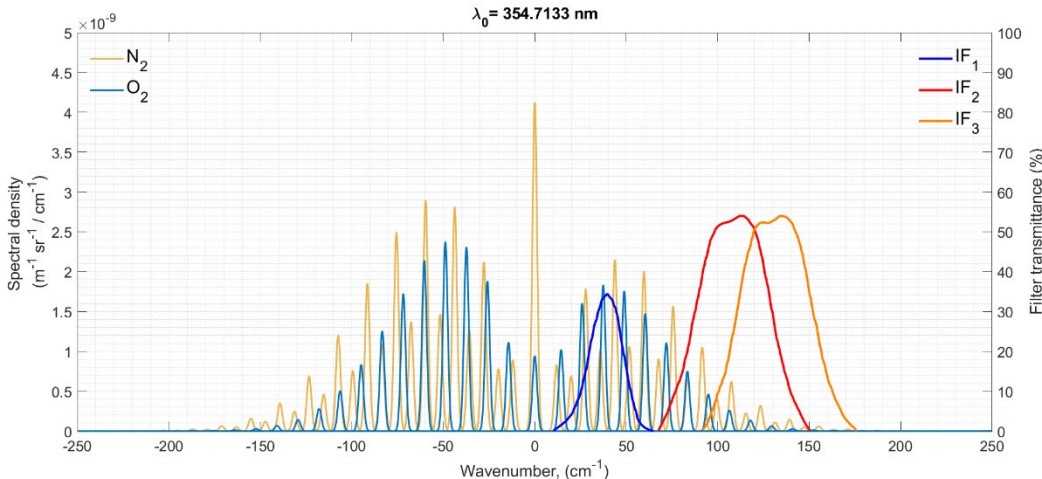

**Figure 1. Cabannes lines and pure rotational backscatter Raman spectrum of $N_2$ (green lines) and $O_2$ (magenta lines) at sea level under the assumption of a Gaussian-widened excitation at 354.7133 nm. The spectrum is given as wavenumber shift from the excitation wavenumber. The transmission curves of the filters and combination of filters in** Hammann et al., 2015 **are superimposed, transposing the wavelength scale in that reference to the wavenumber scale here. The filter denoted as IF2 (red transmission curve) is used in high background-radiation situations, whereas the IF3 filter (orange line) is optimized for low background ones. The Cabannes-line amplitudes (zero shift) are divided by 500 to fit them into the graph.**

**Table 1. Wavelengths and wavenumbers of lasers and filter passbands at half maximum (HM) considered**

|  | Wavelength (nm) | Wavenumber (cm$^{-1}$) | | |
| --- | --- | --- | --- | --- |
| 3$^{rd}$ harmonic of Nd:YAG laser fundamental wavelength | 354.7133 | 28191.78 | | |
| | Non-shifted filters | | Shifted filters* | |
| | HM wavelengths (nm) | HM wavenumbers (cm$^{-1}$) | HM wavelengths (nm) | HM wavenumbers (cm$^{-1}$) |
| Low quantum number filter IF1 | 354.0926 – 354.3488 | 28220.78 – 28241.20 | 353.9852 – 354.2412 | 28229.35 – 28249.77 |
| High quantum number filter IF2 | 353.0767 – 353.6324 | 28277.95 – 28322.46 | 352.9699 – 353.5253 | 28286.52 – 28331.03 |
| High quantum number filter IF3 | 352.7945 – 353.3415 | 28301.23 – 28345.11 | 352.7945 – 353.3415 | 28309.80 – 28353.68 |

* See section 3. A shift of 8.57 cm$^{-1}$ is considered in the filter passbands with respect to those non-shifted.

The power at the filter outputs is shown in Fig. 2, under the assumption of a widened spectrum and of an ideal line spectrum. The effect of the Gaussian spectral widening is so small that the pairs of curves are almost undistinguishable. Likewise, the spectral widening effect is very small in the curves (Fig. 3) obtained as the power ratio between the high quantum number filter outputs (IF2 and IF3 filters in Fig. 1) and the low quantum number filter output (IF1 filter in Fig. 1), henceforth called Q-curves, which, after calibration, give the atmospheric temperature. We conclude that the emission spectral widening of a free-running laser with respect to that of a seeded laser does not have a significant impact on the system performance under

the assumption of a Gaussian widening smaller than 1 cm$^{-1}$ at the fundamental wavelength (Armandillo et al., 1997). No leakage of the Cabannes lines into the filters (especially into the low quantum number one) would be noticed in these conditions, in spite of their higher cross-section (around two orders of magnitude) with respect to the next more intense lines of the rotational Raman spectrum. One should make sure that this hypothesis is satisfied in particular cases and that the presence of aerosols does not increase the return at the excitation wavelength so as to produce leakage.

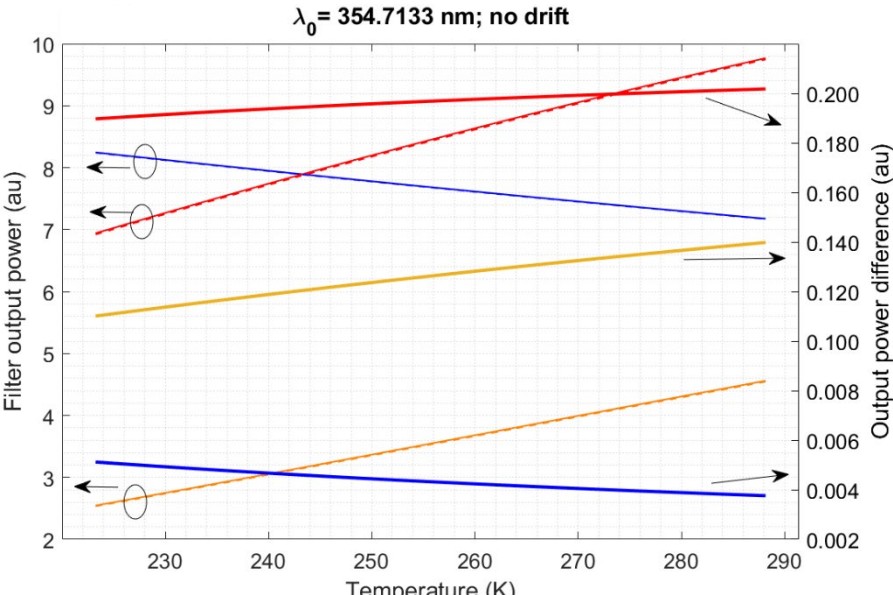

**Figure 2.** Left vertical axis: power (in arbitrary units) at the filter outputs for the widened spectrum (dashed curve) and for an unwidened spectrum (continuous curve) as a function of the atmospheric temperature. The colors of the curves indicate the corresponding filter according to the colors of transmission curves in Fig. 1. There are two curves for each color, but the effect of the Gaussian widening of the spectrum lines is so small that they are almost undistinguishable. Right vertical axis: differences in the filter output power between the unwidened and the widened spectrum. The scale is in the same arbitrary units as the left vertical scale. The colors of the curves indicate the corresponding filter according to the colors of transmission curves in Fig. 1.

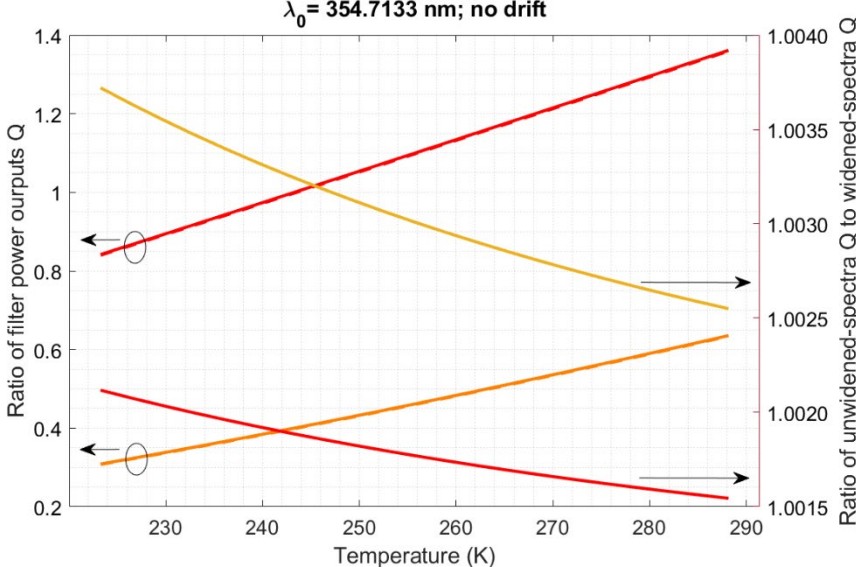

**Figure 3.** Left vertical axis: ratio (Q) between the outputs of the "red" filter and the "blue" filter of Fig. 1 (red lines) and between the outputs of the "orange" filter and the "blue" filter (orange lines) as a function of the atmospheric temperature. The lines corresponding to the widened spectrum (dashed) and those corresponding to the unwidened spectrum (continuous) are almost undistinguishable. Right vertical axis: Ratios of the Q obtained with unwidened spectrum to the Q obtained with widened spectrum. The red curve corresponds to the ratio of Qs between the "red" filter and the "blue" filter, and the orange line to the ratio of Qs between the "orange" filter and the "blue" filter.

## 3 Effect of central wavelength drift

We now turn to the effect of drift of the laser central wavelength. In an unseeded Nd:YAG laser, the central wavelength coincides with the center of the gain curve, which in turn depends on the rod temperature. We have used the approximate analytical models in Sato and Taira, 2012 that reproduce very well experimental results of the central frequency temperature dependence of various Nd:YAG emission lines, among which the one corresponding to the R2 → Y3 transition, in the temperature range 15 ºC – 350 ºC. For the benefit of the reader, we reproduce here Eq. (6) of Sato and Taira, 2012:

$$v_{if}(T) = v_{if}(0) - c_{if}\left(\frac{T}{\Theta_D}\right)^4 \int_0^{\frac{\Theta_D}{T}} \frac{x^3}{e^x - 1} dx \ , \tag{2}$$

where $v_{if}$ is the central wavenumber, $T$ is the temperature, $\Theta_D$ is the Debye temperature, and $c_{if}$ is a fitting parameter. In particular we have taken the transition wavenumber at 0 K as 9403.15 cm$^{-1}$, the temperature-dependence fitting parameter $c_{if}$ as 130.6 cm$^{-1}$, and the Debye temperature as 795 K (Sato and Taira, 2012) (see also Table 1 of this reference). Figure 4

shows the dependence of the center of the 3rd harmonic wavelength emission with the temperature according to that model. The dependence is nearly linear with a slope of approximately 1.5 pm/ºC.

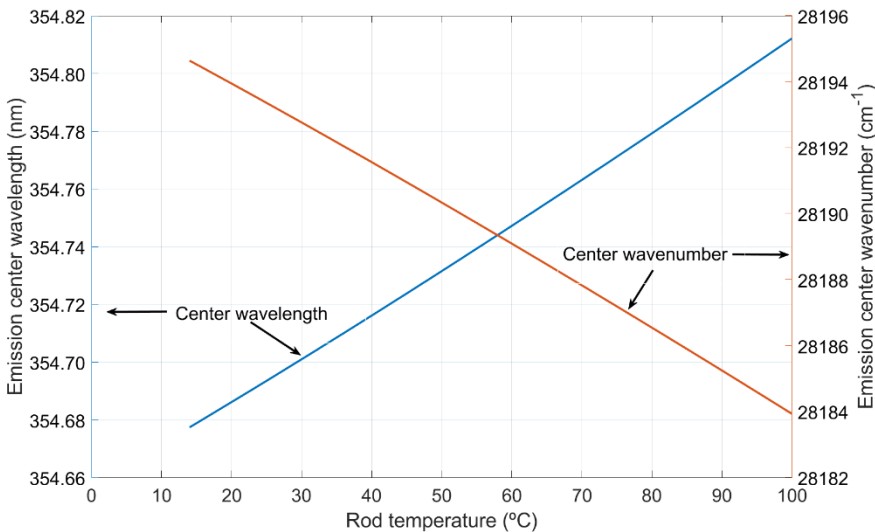

**Figure 4. Dependence of the 3ʳᵈ harmonic emission center wavelength and center wavenumber on the rod temperature.**

To assess the variations of the Q-curve with the laser central wavelength, we assume the central wavelength of 354.7133 nm and the widened spectrum of the unseeded laser already considered in the previous section, and let it drift ±1.5 pm, which would correspond to a temperature variation in the Nd:YAG rod of slightly less than ±1 ºC around 38 ºC. The effect of
the wavelength drift on the power at the filter outputs is shown in Fig. 5, where small differences can be seen (more noticeable in the "red" filter output). However small, these differences impact the Q-curves, as shown in Fig. 6, and imply an uncertainty in the temperature retrieval. For example, let us consider Fig. 7, which is a zoom of the right-hand panel of Fig. 6 around the temperature of 260 K (with the horizontal and vertical axes swapped), where we assume that the Q-curve has been calibrated for a laser rod temperature at 38 °C (solid curve). If the laser central wavelength drifts subsequently by ±1.5
pm, roughly corresponding to a temperature variation of the laser rod of ±1 °C, the uncertainty in the retrieved temperature for the approximately 0.4835 value of Q is the atmospheric temperature interval encompassed by the double-arrowed line, i.e. slightly more than 1 K.

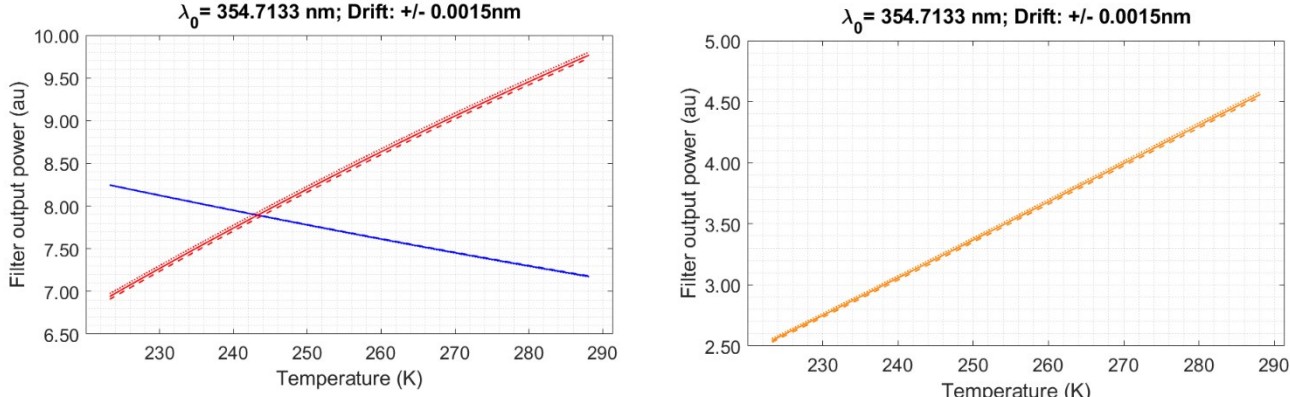

**Figure 5. Effect of the wavelength drift on the power (in relative units) at the filter outputs. Left: IF1 (blue) and IF2 (red) filters of Fig. 1. Right: IF3 (orange) filter of Fig. 1. Solid lines correspond to the filter outputs at the 354.7133 nm wavelength. The dashed line corresponds to the +1.5 pm drift and the dotted one to the -1.5 pm one.**

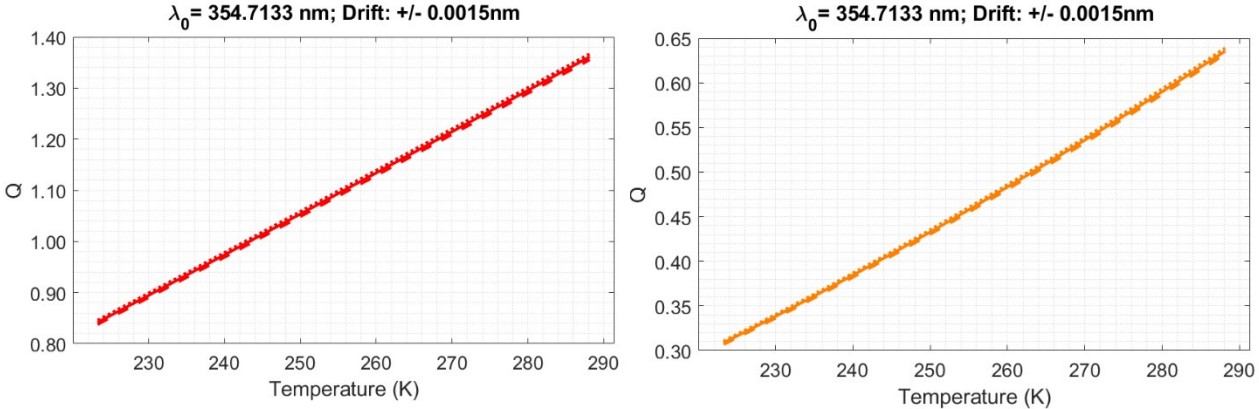

**Figure 6. Effect of the wavelength drift on the Q ratio. Left: ratio between the output power of the IF2 (red) filter and the IF1 (blue) filter of Fig. 1. Right: ratio between the output power of the IF3 (orange) filter and the IF1 (blue) filter of Fig. 1. Solid lines correspond to the filter outputs at the 354.7133 nm wavelength. The dashed line corresponds to the +1.5 pm drift and the dotted one to the -1.5 pm one.**

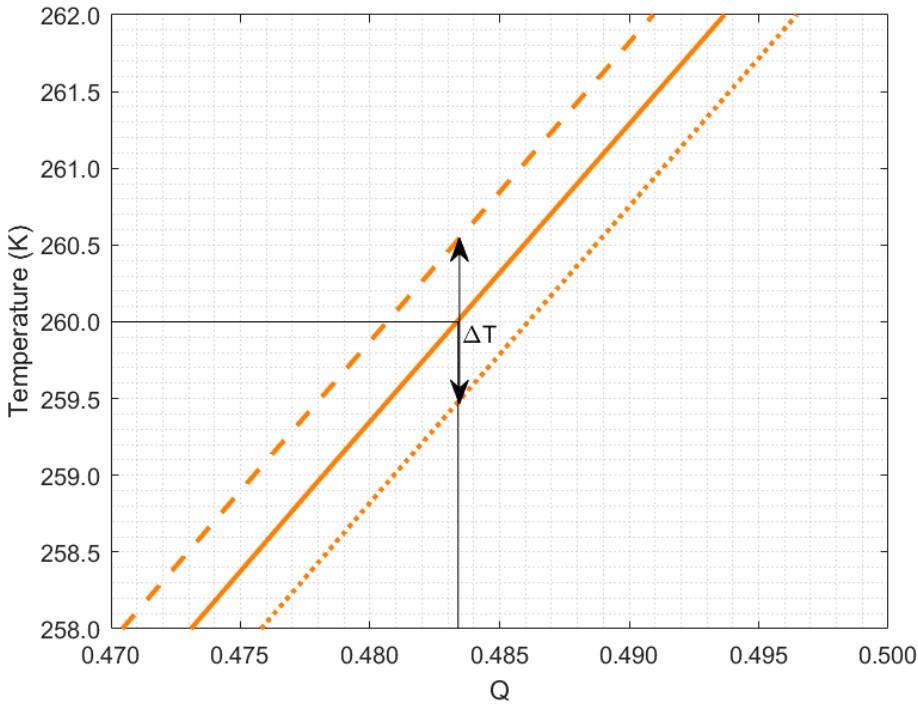

**Figure 7. Example of uncertainty in the temperature retrieval (ΔT) due to the uncertainty in the laser central wavelength. Solid curve: assumed calibrated Q obtained as the ratio between the output powers of the IF3 (orange) filter and the IF1 (blue) filter of Fig. 1 at the 354.7133 nm wavelength; dashed and dotted curves: Q obtained for wavelength deviations with respect to the calibration wavelength of +1.5 pm and -1.5 pm respectively. Note that the axes are swapped with respect to those in Fig. 6 to emphasize that Q is the variable from which the temperature is retrieved.**

This assessment has been done for the laser central wavelength of 354.7133 nm wavelength and a drift of ±1.5 pm around it. However, in Hammann et al., 2015, the laser wavelength was 354.83 nm, fixed by the seeder wavelength. If we assume that the filter responses in Fig. 1 can be shifted by 116.7 pm (i.e. around 9 cm$^{-1}$) towards shorter wavelengths while preserving their shape, in order to be in a similar filter passbands situation with respect to the Raman spectrum as in Hammann et al., 2015, we obtain, for the ratio of IF3 (orange) filter output power to the IF1 (blue) filter output power, the curves of Fig. 8, where the uncertainty in the retrieval of the temperature around 260 K because of a non-controlled ±1.5 pm laser wavelength drift from the calibration wavelength is slightly smaller than 1 K. Note that we have implicitly assumed trailing zeros in the wavelength of the laser in Hammann et al., 2015, which is given only to the hundredth of nm; what is important however is that by shifting the filters by around 9 cm$^{-1}$ we operate in a region where the sensitivity of the Q-curve to wavelength drifts is smaller.

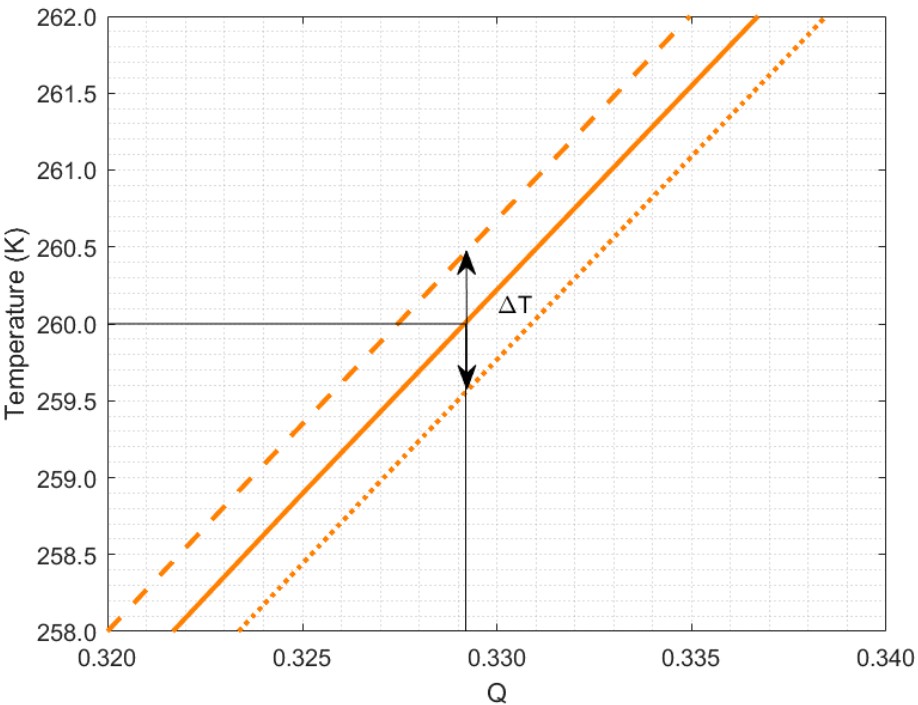

**Figure 8. Like Fig. 7 with the filters of Fig. 2 shifted 8.57 cm⁻¹ towards shorter wavelengths.**

The uncertainty in the temperature retrieval due to a ±1.5 pm wavelength drift with respect to the calibration wavelength has been estimated as in the examples above illustrated in Figs. 7 and 8 for all the considered temperature range, and is shown in Fig. 9 for the case of the filters shifted by 8.57 cm⁻¹. The uncertainty varies between approximately 0.87 K for a 220 K temperature and 1.22 K at 290 K considering that the calibrated Q-curve has been obtained as the ratio of the output power of the IF2 filter of Fig. 1 to the output power of the IF1 filter. For the Q-curve given by the ratio of the output power

of the IF3 filter of Fig. 1 and the IF1 filter, the uncertainty ranges between 0.77 K at 220 K and 1.07 K at 290 K.

## 4 Conclusions

We have estimated, using a real filter configuration, that, in lidar systems employing a Nd:YAG laser to measure the atmospheric temperature profiles through the pure rotational Raman technique, the short-term spectral widening of a commercial-grade unseeded laser virtually does not affect the ratio of the output power of the high quantum-number filter to

the output power of the low quantum-number filter (Q-curves), at least under the assumption that the widening is Gaussian (Armandillo et al., 1997) with a FWHM smaller than 1 cm⁻¹ at the fundamental wavelength. One should pay attention that aerosols do not increase the elastic return to the point that leakage into the filters can happen. For example, the ratio between

the peaks of the Cabannes lines and the peak of the closest $O_2$ line in the ant-Stokes rotational Raman spectrum is about 2000; on the other hand, the backscatter coefficient of dense clouds at 355 nm can typically be up to 500 times the molecular backscatter coefficient at that wavelength, hence a suppression sufficiently higher than $10^6$ (optical density 6) should be achieved at the emission wavelength by the filter to keep the effect of clouds negligible. A suppression of $10^6$ would be sufficient for most aerosol scenarios. However, slight drifts on the central wavelength of the laser emitted spectrum entail small changes in the Q-curves that impair the calibration and cause an uncertainty in the retrieved atmospheric temperature. The drifts can be related to temperature changes of the YAG rod. For the 3rd harmonic of the laser, the drift is around 1.5 pm/°C. The allowable uncertainty in the temperature retrieval depends of course on the application, but if we take as a reference value ±1 K, it turns out that the YAG rod temperature needs also to be kept within ±1 K. Note "en passant" that, in a seeded laser, changes of temperature in the seeder could also cause wavelength drifts, hence uncontrolled biases in the atmospheric temperature measurements that would add to their uncertainty.

Note that, knowing the temperature coefficients of the filters, the analysis could be readily extended to their temperature stability requirements, but this is not the focus of the present study. Likewise, a similar analysis could be carried out for a system working at 532 nm if the filter specifications were known. We have limited the study to an ultra-violet system because the $1/\lambda^4$ law of the Raman cross-section and the lesser background radiation give it an advantage over the visible one, even if the transmitted energy per pulse is lower.

It must be stressed that the analysis above takes only into account the temperature uncertainty due to the spectral behavior of the laser radiation. Although out of the scope of this work, it can be noted that, taking as reference the temperature drifts of the filters quoted in (Johansen et al., 2017), the thermal management of the filter enclosure should be stricter than that of the laser rod. However, as the latter has to cope with heat generated by the pumping system, it is considered more critical. To this, the uncertainty due to the limited signal-to-noise ratio in the detected power should be added (Behrendt and Reichardt, 2000; Di Girolamo et al., 2004).

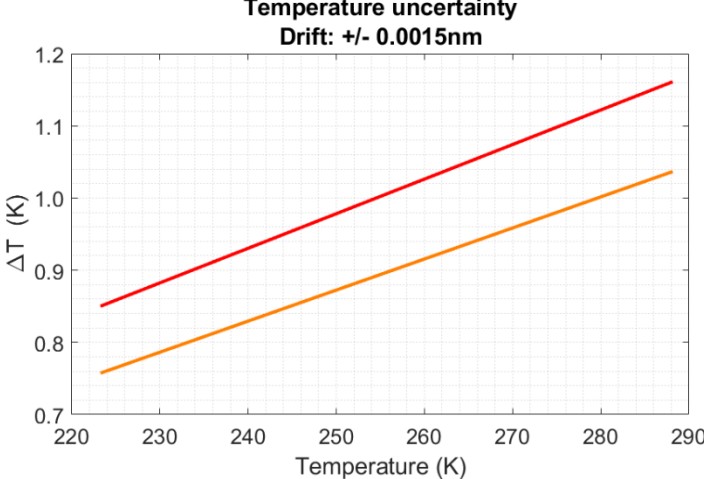

**Figure 9. Retrieved atmospheric temperature uncertainty due to a ±1.5 pm wavelength drift over all the considered temperature range for Q obtained as the ratio between the output powers of the IF2 (red) filter and the IF1 (blue) filter of Fig. 1 (red curve) and as the ratio between the outputs of the IF3 (orange) filter and the IF1 (blue) filter in the same figure (orange curve). The filter responses have been shifted by 8.57 cm⁻¹.**

**Author contribution**

JAZ: formal analysis, software; AC: conceptualization, writing; FD: methodology, formal analysis; AR: visualization, supervision; CM: software, visualization; MS: visualization; NF: validation; AB: supervision, visualization; PDG: supervision, validation.

**Competing interests**

The authors declare that they have no competing interests.

**Acknowledgements**

Author José Alex Zenteno-Hernández thanks CONACyT for the 2018-000068-02NACF-29418 scholarship. The authors with Universitat Politècnica de Catalunya wish to acknowledge the support of the Spain's State Agency for Research (AEI) through the project PID2019-103886RB-I00, of the Catalan Regional Government grant 2021 SGR 01415, and of the European Commission's Directorate-General for Research and Innovation through the Horizon 2020 projects 871115 and 101008004, and through Horizon Europe project 101086690. The contribution of Paolo Di Girolamo and Noemi Franco to

this paper was possible thanks to the support from the Italian Ministry for Education, University and Research under the Grants OT4CLIMA and FISR2019-CONCERNING, the support of the Italian Space Agency under the Grant As-ATLAS and CALIGOLA and the support of the European Commission under the Grant FESR 2014-2022 "STAC-UP".

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
