# Peer review of "On the temperature stability requirements of free-running Nd:YAG lasers for atmospheric temperature profiling through the rotational Raman technique"

_Atmospheric Measurement Techniques, 2024_

## Referee Comment (RC2)

**1    General comment**

Review of the manuscript "On the temperature stability requirements of free-running Nd:YAG lasers for atmospheric temperature profiling through the rotational Raman technique", by José Alex Zenteno-Hernández, Adolfo Comerón, Federico Dios, Alejandro Rodríguez-Gómez, Constantino Muñoz-Porcar, Michaël Sicard, Noemi Franco, Andreas Behrendt, Paolo Di Girolamo.
The presented study provides quantitative numbers about the uncertainty in the calculated temperature as a result of a Gaussian widening of the emitted line and the resulting impact on the detected high and low number lines and their ratio Q. It also simulates the effect of a drift in the central emitted line assessing the impact on the power ratio Q. The assessment is done for a free running Nd:YAG laser (unseeded) with respect to a seeded laser and shows that the overall uncertainty due to the changing temperature of the rod can be kept around 1K.

**2    Specific comments**

The manuscript provides a lean, concise, but clear technical description of the uncertainty sources related to the instability of key parameters of the emission laser. It is probably more of a technical note than a scientific paper, but I believe that many authors working in the Raman lidar filed and retrieving the temperature from the PRR spectrum will benefit from this article when calculating their global error budget.

**3    Technical corrections**

The overall quality of the graphics should be improved, especially Fig.2-7 something is always missing or not properly shown. The list of detailed comments is given here:

- Page 1 Line 22 add a space before the bracket.

- Page 1 line 28: "the backscatter cross section corresponding to the Raman lines..."

- Page 2 Line 41: do the author mean the thermal coefficient of the interference filter? Shouldn't be "ppm/°C?

- Page 2 Line 47-50: something is missing, please check and consider rephrasing this paragraph.

- Line 65 and 67: remove brackets from Hammann's citation.

- all over through the text: replace atmosphere temperature with atmospheric temperature.

- Figures 2 and 3 are not very useful in the way they are shown. A plot showing the difference-curve of "widened-unwidened", in y-log scale and with a legend would surely help.

- Line 96: in spite of.

- Line 112 (and all over through the text): remove brackets around the authors' names.

---

## Author Comment (AC1)

We thank the reviewers for their thoughtful remarks, which allow us to improve our manuscript. In the following we address the remarks in detail.

**Responses to reviewer 1**

General technical comment: The overall quality of the graphics should be improved, especially Fig.2-7 something is always missing or not properly shown

**Answer:** Figs. 2-8 have been revised according to the comments of reviewer 2 and also to correct some small calculation inconsistencies detected after the submission.

**Detailed comments:**

**Comment 1**. Page 1 Line 22 add a space before the bracket.

*Answer*: Done.

**Comment 2**. Page 1 line 28: "the backscatter cross section corresponding to the Raman lines..."

*Answer*: Done.

**Comment 3**. Page 2 Line 41: do the author mean the thermal coefficient of the interference filter? Shouldn't be "ppm/°C?

*Answer*: We mean the thermal dependence of the central wavelength in picometers (pm)/°C.

**Comment 4**. Page 2 Line 47-50: something is missing, please check and consider rephrasing this paragraph.

*Answer*: We think that the paragraph is complete. However, it is true that it is too long. We have broken it into shorter sentences with some rephrasing. Now it reads:

"In this paper we assume that the receiver is in a well-controlled environment, such that the wavelength drift of the filters can be neglected. We focus henceforth on the wavelength stability requirements for a Nd:YAG non-seeded laser to be used in a lidar measuring atmospheric temperature profiles using the pure rotational Raman spectra of $N_2$ and $O_2$ under the excitation by the third harmonic of the laser fundamental frequency".

**Comment 5**. Line 65 and 67: remove brackets from Hammann's citation.

*Answer*: Done.

**Comment 6**. all over through the text: replace atmosphere temperature with atmospheric temperature.

*Answer*: Done.

**Comment 7**. Figures 2 and 3 are not very useful in the way they are shown. A plot showing the difference-curve of "widened-unwidened", in y-log scale and with a legend would surely help.

*Answer*: Fig. 2 has been modified adding a right y-axis showing the difference between the "unwidened" and "widened" curves. Likewise, a right y-axis has also been added to fig. 3 to show the ratio between "widened" and "unwidened" Qs.

**Comment 8**. Line 96: in spite of.

*Answer*: Done.

6. Line 112 (and all over through the text): remove brackets around the authors' names.

***Answer***: Done.

---

## Author Comment (AC2)

We thank the reviewers for their thoughtful remarks, which allow us to improve our manuscript. In the following we address the remarks in detail.

**Responses to reviewer 2**

**Major comment 1:** The prime concern I have is that the authors seem to have picked a confined problem and analyzed it, but that that analysis is somewhat unportable. It seems that the results are specific to a set of interference filters that have a known center location and full width at half maximum. Most Raman lidar systems have slight variations in this parameter and thus the broader applicability is reduced. Not being able to broaden the results from the single system analysis seems to dramatically limit the impacts of this work

*Answer***:** It is true that the results presented are somewhat limited in that they apply to a particular setup. However, the general characteristics of the filters considered are the result of a deep trade-off study carried out by Radlach et al. (M. Radlach, A. Behrendt and V. Wulfmeyer, "Scanning rotational Raman lidar at 355nm for the measurement of tropospheric temperature fields", 2008) and refined in manuscript's ref. Hammann et al., 2015. In a way or another, it is expected that modern temperature lidars use filters (perhaps implemented according to different techniques) with characteristics not deviating much of those considered in the present study. For completeness, we have added Radlach's reference to the manuscript. Additionally, while the specific numbers may sligthly vary in case of selection of filters with different spectral specifications, the general conclusions of the manuscript (i.e. (i) the **short-term spectral widening** of a commercial-grade unseeded laser virtually **does not affect** the power ratio of the high quantum-number over the low quantum-number signal (**Q-curves**) and ii) the **Nd:YAG rod temperature** should also be kept **within 1 K** in order to keep the **bias** affecting atmospheric temperature measurements **smaller than 1 K**) **are in no extent affected by these slight changes and keep their validity**.

**Major comment 1.1.** What about different passband filters? If you took the same filter center wavelengths and changed the width of the filters, are your results (i.e. the finding that you need approximately 1C temperature stability in your laser rod) robust?

*Answer:* See reasoning above. The following sentences have been added shortly after Eq. (1):

"These filters were specified after a thorough analysis of the system trade-off between sensitivity to temperature difference and statistical-noise induced uncertainty (Radlach et al., 2008; Hammann et al., 2015b); for this reason, we expect their characteristics (bandwidth and central-wavelength distance to the laser emission line) be representative of modern Raman temperature lidars using interference filters". Once again, even in case of selection of filters with different spectral specifications in terms of both filter width or central wavelength, the above recalled general conclusions of the manuscript **are in no extent affected.**

**Major comment 1.2.** What about filters are different center wavelengths? If you took the same filter set and simply tilted them, their wavelengths would shift to the blue. Alternatively, if you picked different center wavelengths for different temperature sensitivity, the analysis here would need to be altered. Again are your results robust with reasonably different center wavelengths?

*Answer*: We think that the sentences included as a response to the previous remark cover this one as well. Note as well that an equivalent case (a shift of the emitted wavelength) is indirectly considered in the paper (penultimate paragraph of section 3), when the emission wavelength is assumed to pass from 354.7133 nm to 354.8300 nm. The uncertainty in the temperature retrieval due to a $\pm 1.5$ pm laser central wavelength drift changes, but not dramatically, as it stays at around 1K. Once again, even in case of selection of filters with different spectral specifications in terms of both filter width or central wavelength, the above recalled general conclusions of the manuscript **are in no extent affected.**

**Major comment 1.3.** What does the laser output wavelength do to the Cabannes line and at what width do you start to worry about contamination into your filters? Said differently, if you use the same filters and start to alter the laser rod temperature (or lack control enough that the Cabannes line widens significantly), at what point does the issue become lack of stability of the laser and more that the contamination for elastic scattering is unavoidable?

*Answer*: We are not sure to understand completely the reviewer's remark. As the reviewer points out in his/her general comments, "The laser output spectrum is convolved with the ideal Raman spectrum and passed through ideal optical filters and the difference between calibration curves of the seeded and unseeded lasers are compared". Therefore, the Cabannes lines has the same Gaussian shape and width as the emitted radiation. With the output spectrum central wavelength (354.7133 nm) and width (3 cm$^{-1}$ full-width half-maximum), the spectral density of the $N_2$ backscatter coefficient for the Cabannes line is about 10 orders of magnitude less than the backscatter coefficient peak spectral density of the closest line (an $O_2$ line) of the anti-Stokes rotational Raman spectrum. The shape of the aerosol return spectrum is also going to be the convolution of an ideal delta-like backscatter coefficient with the laser (Gaussian, 3-cm$^{-1}$ wide) spectrum, so the aerosol backscatter coefficient should be about 10 orders of magnitude higher than the molecular backscatter coefficient (for the most common aerosol scenarios the aerosol and the molecular backscatter spectrum are of the same order of magnitude at 355 nm) for the elastic-return spectral density to equal the peak spectral density of the closest line in the rotational Raman spectrum, which is not physically reasonable. The real problem would be an insufficient suppression of the elastic return wavelength by the low quantum-number filter. There is about a $2 \times 10^3$ ratio between the peak of the Cabannes lines and the peak of the closest $O_2$ line of the anti-Stokes rotational spectrum. On the other hand, the backscatter coefficient of dense clouds at 355 nm can typically be up to 500 times the molecular backscatter coefficient at that wavelength, hence a suppression sufficiently higher than $10^6$ (optical density 6) should be achieved by the filter at the emission wavelength to keep the effect of clouds negligible. A suppression of $10^6$ would be sufficient for most aerosol scenarios. To make this point clearer, we have added the following explanation right after the sentence "One should pay attention that aerosols do not increase the elastic return to the point that leakage into the filters can happen" on lines 184-185 of the original manuscript:

"For example, the ratio between the peaks of the Cabannes lines and the peak of the closest $O_2$ line in the ant-Stokes rotational Raman spectrum is about 2000; on the other hand, the backscatter coefficient of dense clouds at 355 nm can typically be up to 500 times the molecular backscatter coefficient at that wavelength, hence a suppression sufficiently higher than $10^6$ (optical density 6) should be achieved at the emission wavelength by the filter to keep the effect of clouds negligible. A suppression of $10^6$ would be sufficient for most aerosol scenarios"

**Major comment 1.4.** What about 532nm vs 355nm? Because all the analysis is basically done in cm-1, it seems like a relatively simple exercise to, at minimum, comment on the same style Raman lidar system using the 2nd harmonic of an Nd:YAG laser.

*Answer*: It is true that, knowing the filter characteristics of a 532 nm system, a completely similar analysis could be carried out. However, we have chosen a 355 nm system because we had a complete characterization of the filters and, in addition, the Raman cross-section is approximately 5 times higher at 355 nm than at 532 nm, which in general offsets the lower emitted power. Moreover, the background radiation (although not addressed in the paper) is also lower than at 532 nm. This is explained in the following paragraph, added after line 189 of the original manuscript, which also addresses the later remark on the temperature coefficient of the filters:

"Note that, knowing the temperature coefficients of the filters, the analysis could be readily extended to their temperature stability requirements, but this is not the focus of the present study. Likewise, a similar analysis could be carried out for a system working at 532 nm if the filter specifications were known. We have limited the study to an ultra-violet system because the $1/\lambda^4$ law of the Raman cross-section and the lesser background radiation give it an advantage over the visible one, even if the transmitted energy per pulse is lower"

**Major comment:** In general, I am imagining a set of contour figures that would allow the filter properties to change slightly, where the contour is the maximum temperature deviation allowable for the laser rod for a set observational temperature error and the axes are widths or center wavelengths of each filter.

*Answer:* This kind of contours are generated by Radlach et al., 2008 and by Hamman et al., 2016 to optimize the system performance against noise. We have preferred not to depart from that optimized situation and just analyze what it implies for the laser wavelength stability. See answer to Major comment 1.1.

**Major comment 2:** The subject nature of this paper somewhat naturally uses different units that mean basically the same thing. However, the authors jump between the two rather freely, which is somewhat difficult to follow in my opinion.

*Answer*: see responses to the following detailed comments

**Major comment 2.1**: For units used to described wavelengths and wavelength changes, I would show both set of units (cm-1 and nm) simultaneously to aide the reader that is not comfortable jumping between the two. For example, I tend to think of laser spectral output characteristics in units of MHz. This is well done in Table 1 but could be replicated in the text.

*Answer*: done

**Major comment 2.2**: Same comment for Figure 4.

*Answer*: Fig. 4 has been modified according to this remark.

**Major comment 2.3**: I see both Kelvin and Celsius (lines 185-186 for example). I would pick one and stick with it.

*Answer*: Because two different temperatures are referred to in the paper (the atmospheric temperature and the laser rod temperature), to avoid confusion between both we have chosen to use K for the atmospheric temperature and Celsius for the rod temperature, hoping that this will help the reader to interpret which temperature we are dealing with. We have made sure that this convention is consistently held throughout the text.

**Major comment 3**: Figures 2/3/5/6: I note several issues with these figures:

**Major comment 3.1:** You specify that there are plotted widened and unwidened lines in Figures 2 and 3 but never specify which is dashed and which is solid.

**Major comment 3.2:** I would move your title to the y-axis as that better illustrates what you are plotting and the units of that.

**Major comment 3.3**: Using two lines that are almost on top of one another is not really a clear way to present your data. I would suggest using a second y-axis to plot the difference for all four figures. If you choose to do something other, please make sure that it is clear and easy for the interested reader to understand the magnitude of the changes between solid and dashed lines.

*Answer to major comment 3 and its associated specific questions*: The figures have been modified following the reviewer's recommendations.

**Major comment 4:** if I am reading your conclusions correctly, the temperature sensitivity of the interference filters (2-5 pm/C) are at least 33% higher than the laser rod (1.5 pm/C). I recognize that the two will have different thermal management systems, but it does seem like a point that needs to be touched upon. You never mention why you are focused on the laser rod when, ostensibly, the filters are more sensitive.

*Answer*: See answer to major comment 1.4. In addition, the following sentences have been added after the period in line 191 of the original manuscript:

"Although out of the scope of this work, it can be noted that, taking as reference the temperature drifts of the filters quoted in (Johansen et al., 2017), the thermal management of the filter enclosure should be stricter than that of the laser rod. However, as the latter has to cope with heat generated by the pumping system, it is considered more critical"

**Minor Comments:**

1. Introduction: The authors never really say why one should care about their conclusions. I assume this is because lidar designs that lack a seed laser are cheaper, and if you can get similar performance with cheaper lasers, that makes the technique more useful. However, I suggest the authors state the motivation for using unseeded lasers directly.

*Answer*: we have added this sentence to the introduction, at the end of line 46 in the original manuscript:

"As the cost of a free-running laser is much lower than that of a seeded laser for equivalent output energies, the use of the former can be of advantage provided the cost of achieving the wavelength stability assessed in the present paper does not offset the reduction on laser cost."

2. Lines 23-24: The term "continuous" here requires context. I assume the authors mean on the scale of a few minutes. However, the statement reads (in my opinion) like radiosondes can only be launched during campaigns, which is obviously not true. I suggest adding a numerical definition for the temporal resolution required to be "continuous".

*Answer*: The sentence in lines 23-24 has been changed to:

"nevertheless, out of specific measurement campaigns, radiosondes are launched only every 12 hours from most meteorological-service sites; on the other hand, Raman lidars can operate on a 24/7 basis, obtaining time resolutions in the range from minutes to a few hours. Therefore,"

3. Line 29: "high quantum numbers" should specify "rotational quantum numbers".

*Answer*: we have added "rotational" in line 29 of the original manuscript.

4. Line 68: It is probably worth noting that the wavelengths you specify here are reference to vacuum wavelengths.

*Answer*: added in line 68.

5. Figure 1: This figure will be difficult to read for those that are colorblind because you have used both red and green. Suggest modifying colors or including different line styles to be easier to differentiate for such people.

*Answer*: done our best to improve the visibility of this figure with a change of colors and increasing the line thicknesses. Using different types of line resulted in a confusing graph in the zones where several Gaussian curves overlap.

6. Figure 1: The Cabannes line is not part of the pure rotational Raman spectrum because it has no Raman shift. I would either remove the line or not call the presented spectrum the "Pure rotational backscatter Raman spectrum".

*Answer*: The text and the caption of fig.1 have been modified to differentiate the Cabannes lines from the pure rotational Raman spectrum.

7. Table 1: I am not seeing where "HM" is defined.

*Answer*: The caption of Table 1 now includes the definition of HM.

8. Lines 94-97: This statement seems experimental in nature, whereas the rest of the manuscript is theoretical. If these are real test results, to what system do they apply?

*Answer*: This statement is theoretical, based on the data of the responses of the filters of the reference system described in Hammann et al 2015. To make it clearer, we have changed the sentence starting in line 94 of the original manuscript: "No leakage […] is noticed…" to "No leakage […] would be noticed…"

9. Line 117: The contents of this line are really extensions of the previous line and should not be a separate paragraph.

*Answer*: The paragraph structure has been remodeled according to this comment.

10. Figure 7/8: I see no clear reason to flip the x and y axis? If there is a good reason, please tell the reader why it is helpful. If not, I would flip it back for consistency.

*Answer*: Respectfully, we prefer to keep figures 7 and 8 as they provide the final result of the analysis. We have added an explanation in the caption of figure 7:

"to emphasize that Q is the variable from which the temperature is retrieved"

**Comments about References:**

1. Line 36: The reference to Hammann et al 2015 is not a seminal reference for calibration of Raman lidar by radiosonde. I would either use an "e.g." or pick a more original reference.

*Answer*: we have included "e.g." in the text.

2. Line 60: In my reading, the citation to Armandillo et al 1997 is not terribly appropriate. The laser presented by Armandillo has a nearly Gaussian profile. This citation seems to indicate however that the Armandillo manuscript is used to suggest that Nd:YAG lasers must have Gaussian profiles. I would find a more fundamental reference here as this assumption is pretty important.

*Answer*: We do not say that Nd:YAG pulsed lasers have Gaussian emission spectra shot by shot, but that the averaged spectrum over many pulses is, at least to a good degree of approximation, Gaussian. This is the conclusion of Armandillo's et al. paper for the pulsed laser they have characterized (fig. 1 in Armadillo's et al., 1997 is the spectral profile averaged over 1000 shots). To the best of our knowledge, Armandillo et al 1997 is the only reference that measures the statistical emission spectrum of a free-running Q-switched Nd:YAG. We have been unable to find a more fundamental reference giving a theoretical basis for it.

3. Line 70: You have said that the spectrum frequencies from Zenteno-Hernández et al. 2021 relies on previous work from itself. I would remove the second reference at the end of the line.

*Answer*: Done. Thank you for detecting this inconsistency.

**Typographic Comments:**

1. Line 25: "spending" is probably not the word you are looking for here. Perhaps "launching"?

*Answer*: We think that "spending" in the sense of "to use up" or even "to consume wastefully" (Merriam-Webster online dictionary) is appropriate. However, we have replaced it by "expending", with the more usual meaning of "to make use of for a specific purpose".

2. Line 29: The antecedent of "has" is "lines". This should read "…while that of lines with high [rotational] quantum numbers have…"

*Answer*: Respectfully, we think the antecedent of "has" is "that" (referring to the "backscatter cross-section"). We have added "rotational" before quantum numbers.

3. Line 71: There is an extra ")" after Buldakov et al. 1996 that should be removed.

*Answer*: Done.

4. Line 92: "atmosphere temperature" should be "atmospheric temperature".

*Answer*: Corrected.

5. Line 96: "…in spite their higher…" should be "…in spite of their higher…"

***Answer***: Done.

6. Line 128: "entrain" is an odd choice of word in my opinion. I would suggest swapping with "include" or "imply".

***Answer***: Thanks for the suggestion. We have changed it to "imply".

---

## Author Response (AR2)

The issues raised by reviewer #2 have been addressed. The reference list has also been checked, one repeated reference has been removed and wo other references have been completed.